# Multiple ETS Factors Participate in the Transcriptional Control of *TERT* Mutant Promoter in Thyroid Cancers

**DOI:** 10.3390/cancers14020357

**Published:** 2022-01-12

**Authors:** Caitlin E. M. Thornton, Jingzhu Hao, Prasanna P. Tamarapu, Iñigo Landa

**Affiliations:** 1Division of Endocrinology, Diabetes and Hypertension, Department of Medicine, Brigham and Women’s Hospital, Boston, MA 02115, USA; cthornton3@bwh.harvard.edu (C.E.M.T.); jhao3@bwh.harvard.edu (J.H.); 2Harvard Medical School, Boston, MA 02115, USA; 3Human Oncology and Pathogenesis Program, Memorial Sloan Kettering Cancer Center, New York, NY 10065, USA; prasanna826@gmail.com

**Keywords:** *TERT* promoter mutations, ETS factors, thyroid cancer, transcriptional regulation, MAPK signaling

## Abstract

**Simple Summary:**

Mutations in the promoter region of the telomerase reverse transcriptase (TERT) gene are enriched in patients with advanced thyroid tumors. Their consequence is the reactivation of TERT expression through mechanisms that involve specific transcription factors and other signaling inputs. Here, we show that, contrary to what it has been shown in other tumor types, multiple factors of the ETS family are able to control TERT transcription and they do so in both the presence and absence of promoter mutations. We also show that TERT expression is more dependent on the MAPK signaling pathway in thyroid cells without *TERT* promoter mutations. Our work points to an intricate, still not fully characterized, regulatory network of TERT transcription in thyroid tumors. We caution against the assumption of mechanisms identified in other cancer lineages being identical in *TERT*-mutant thyroid specimens.

**Abstract:**

Hotspot mutations in the *TERT* (telomerase reverse transcriptase) gene are key determinants of thyroid cancer progression. *TERT* promoter mutations (TPM) create de novo consensus binding sites for the ETS (“E26 transformation specific”) family of transcription factors. In this study, we systematically knocked down each of the 20 ETS factors expressed in thyroid tumors and screened their effects on TERT expression in seven thyroid cancer cell lines with defined TPM status. We observed that, unlike in other TPM-carrying cancers such as glioblastomas, ETS factor GABPA does not unambiguously regulate transcription from the *TERT* mutant promoter in thyroid specimens. In fact, multiple members of the ETS family impact *TERT* expression, and they typically do so in a mutation-independent manner. In addition, we observe that partial inhibition of MAPK, a central pathway in thyroid cancer transformation, is more effective at suppressing *TERT* transcription in the absence of TPMs. Taken together, our results show a more complex scenario of *TERT* regulation in thyroid cancers compared with other lineages and suggest that compensatory mechanisms by ETS and other regulators likely exist and advocate for the need for a more comprehensive understanding of the mechanisms of *TERT* deregulation in thyroid tumors before eventually exploring TPM-specific therapeutic strategies.

## 1. Introduction

Mutations in the proximal promoter of *TERT* (telomerase reverse transcriptase) are the first genomic lesions in a gene regulatory region and display a high prevalence across multiple cancer types [1]. They were first identified in cutaneous melanoma [2,3] and subsequently reported in numerous malignancies [4], including thyroid carcinomas [5,6]. *TERT* promoter mutations (TPMs) are observed with increasing frequency according to the severity of thyroid cancer [7,8]. Thus, only 9% of the papillary thyroid cancers (PTCa), a predominant subtype with overwhelmingly good prognosis, harbor TPMs, and these are typically subclonal within the tumor [9]. In contrast, we reported that TPM prevalence is much higher in advanced thyroid tumors, which account for most of the morbidity and mortality of the disease: they occur in 40% of poorly differentiated (PDTC) and 73% of anaplastic thyroid cancers (ATC) [8]. Within each thyroid tumor subtype, TPMs associate with more aggressive features, particularly in combination with constitutive activation of the MAPK (“mitogen-activated protein kinase”) pathway, usually via BRAF^V600E^ driver mutation [8,10,11]. Overall, this stepwise increase in frequency implies that clonal TPMs confer a strong selective advantage as PTC progresses to PDTC/ATC.

TPMs occur at either c.-124C>T or c.-146C>T in a mutually exclusive fashion. These single nucleotide substitutions generate an identical 11-nucleotide stretch creating de novo binding sites for the ETS (“E26 transformation specific”) family of transcription factors, which comprise 27 members with diverse functions and tissue-specific roles [12]. As a result, *TERT* transcription, which is typically silenced in adult tissues, is reactivated in cells carrying TPMs. The seminal papers reporting TPMs in melanoma pointed at an ETS-mediated role on *TERT* mutant promoter control [2,3], but the identification of the specific factors involved in this regulation, particularly in thyroid cancers, has proven not to be straightforward. Nevertheless, distinguishing which ETS (and other) factors are important in the reactivation of *TERT* in aggressive thyroid tumors is a necessary first step towards enabling new methods of blocking *TERT* expression and ultimately inhibiting cell growth in a cancer-specific manner.

Several studies in *TERT*-mutant glioblastomas demonstrated that GABPA, an ETS factor acting as a tetrameric protein along with its GABPB isoform, drives *TERT*-mutant transcription in this tumor lineage, opening the door to targeting the GABPA/GABPB axis to achieve *TERT* downregulation in that context [13,14,15]. The former prompted the *a priori* attractive idea of GABPA also being the key player in *TERT*-mutant regulation across TPM-harboring thyroid cancers [16]. However, several other ETS members, including ELK1 and ETV1/4/5, have been proposed to control *TERT* transcription in thyroid specimens [11,17,18], suggesting that promiscuity and/or compensatory mechanisms among ETS and possibly other proteins exist. A recent study showed that GABPA has tumor suppressor properties in thyroid tumors, rendering this factor an unadvisable target for inhibition in this lineage [19]. In addition, some ETS proteins as well as other MAPK-mediated factors such as Sp1 and c-Myc are also able to bind and control the *TERT* wildtype promoter [20,21,22,23].

Thus far, most thyroid cancer studies aimed at identifying mechanistic underpinnings of *TERT* mutant control have either focused on single ETS factors in several cell lines or screened multiple factors in a limited number of lines. Here, we adopted an agnostic approach, screening all ETS factors that are expressed in thyroid tumors in numerous, fully characterized, thyroid cancer cell lines. Our results point to a greater complexity of TPM-mediated regulation of *TERT* transcription from what was previously reported and caution against adopting what has been demonstrated in other cancer types carrying TPMs into thyroid tumors. Furthermore, we observed that the MAPK blockade was more efficient at suppressing *TERT* transcription in cells without TPMs, suggesting that a switch in *TERT*-mutant control might operate in thyroid cancers carrying TPMs. Overall, we believe that only observations that are generalizable to all thyroid cancers harboring TPMs will help in exploiting them as potential targets to implement tumor-specific epigenetic inhibition of this *bona fide* cancer gene.

## 2. Materials and Methods

### 2.1. Analysis of Public Expression Datasets

Normalized transcriptomic data from The Cancer Genome Atlas (TCGA) evaluation of PTC [9] was either downloaded from publicly available repositories (http://firebrowse.org/?cohort=THCA&download_dialog=true, accessed on 1 January 2018) or retrieved from our published studies on PDTC, ATC, and thyroid cancer cell lines [8,24]. Expression data for the specific 27 factors from the ETS family were recovered from these datasets and plotted accordingly.

### 2.2. Cell Culture

The following authenticated thyroid cancer cell lines, which we recently characterized, were used in this study [24]. Cell lines were cultured in RPMI (Cal62, MDAT41, CUTC5, BCPAP, SW1736, and T238) or DMEM (TCO-1, Hth7, K1, and U251) medium, supplemented with 10% FBS, penicillin, streptomycin, and L-glutamine. BHT101 cells were cultured in MEM with 20% FBS, penicillin, and streptomycin. All cell lines were passaged at approximately 80% confluency and maintained in a 37 °C humidified incubator with 5% CO_2_.

### 2.3. Gene Silencing

We evaluated the effect of ETS silencing on *TERT* transcription via stable gene silencing. We individually silenced each of the 20 ETS factors that are expressed in thyroid cancers. To this end, we used the TRC collection of hairpins from the Genetic Perturbation Platform at the Broad Institute (https://portals.broadinstitute.org/gpp/public/, accessed on 1 February 2018). Short hairpin RNA (shRNA) clones were obtained from the MSKCC Gene Editing and Screening core in either the pLKO.1 or pLKO_TRCN005 plasmid backbones. The specific shRNA clones and target sequences used in our study are listed in Appendix A. Briefly, bacterial glycerol stocks were grown and purified using EndoFree Plasmid Kits (Qiagen, Hilden, Germany). Lentiviral production was performed co-transfecting 293-FT packaging cells with individual pLKO_shETS plasmids, psPAX2 (packaging plasmid), and pMD2.G (envelope plasmid) using FuGENE (Promega, Madison, WI, USA) or Lipofectamine3000 (ThermoFisher Scientific, Waltham, MA, USA) and following standard protocols. The target cells were subsequently infected with these lentiviruses and selected under 1 ug/mL of puromycin for at least one week. A total of 22 shRNAs against ETS factors were used: one for each of the 20 ETS factors, with the exception of ETS2 and GABPA, for which a mixture of equal amounts of two hairpins targeting different regions of the same gene achieved better silencing than individual hairpins (see Appendix A).

### 2.4. Real-Time Quantitative PCR

Cells were lysed using TRIzol reagent, and RNA was isolated with chloroform and then precipitated from the aqueous phase using isopropanol. cDNA was synthesized using the High Capacity cDNA Reverse Transcription Kit (Applied Biosystems, Waltham, MA, USA) according to the manufacturer’s instructions. Quantitative PCR was performed on a Roche LightCycler^®^ 480 II. SYBR™ Select Master Mix (Applied Biosystems) for measuring actin and ETS factors. TaqMan™ Gene Expression Master Mix (Applied Biosystems) and *TERT* TaqMan Assay (Applied Biosystems, Hs00972650_m1) were used to measure *TERT* expression. Cells infected with pLKO_shScramble constructs were used as reference for each experiment. Specific forward and reverse primers used for these reactions are available in Appendix A. The expressions of ETS and *TERT* were normalized to the beta-actin (*ACTB*) housekeeping gene using the delta-delta Ct method.

### 2.5. MAPK Inhibition Experiments

We ran MAPK inhibition experiments using the FDA-approved MEK inhibitor trametinib and RAF inhibitor dabrafenib (Selleck Chemicals, Houston, TX, USA). We calculated IC_50_ values for each cell line to account for inter-specimen variability to MAPK inhibition and to make our comparisons accurate. For IC_50_ calculations, 50,000 cells were seeded in 6-well plates, six conditions per treatment in triplicates (trametinib: 0 nM (DMSO), 1 nM, 2 nM, 5 nM, 10 nM, and 30 nM; dabrafenib: 0 nM, 5 nM, 10 nM, 20 nM, 40 nM, and 80 nM). Cells were counted after incubation for 72 h, and IC_50_ values were calculated using linear regression. For the MAPK inhibition treatment experiments, cells were seeded using media with 1% FBS into 4 cm dishes at a density of 4 × 10^5^ cells per plate. After 24 h, the cells were treated with trametinib only (Cal62, KRAS^G12D^-mutant) or trametinib plus dabrafenib (MDAT41, BCPAP, and BHT101; all three BRAF^V600E^-mutant) at the IC_50_ dose determined for each cell line. After 6 h, the cells were harvested in total lysis buffer with protease and phosphatase inhibitors (TRIS pH 7.4 10 mM; NaCl 50 mM; MgCl_2_ 2 mM; 1% SDS; and Sigma P5726, P0044, and P8340) for protein analysis and in TRIzol for RNA isolation.

### 2.6. Western Blotting

Total protein concentrations were measured by Pierce BCA (ThermoFisher Scientific). Equal amounts of proteins were run on NuPAGE 4 to 12%, Bis-Tris protein gels (ThermoFisher Scientific) and subsequently transferred onto PVDF membranes (ThermoFisher Scientific) by wet electroblotting. Transfer efficiency was checked with Ponceau staining. MAPK inhibition was evaluated via a phospho-ERK antibody (Cell Signaling Technology, #4292, Danvers, MA, USA). Equal protein loading was controlled with the p85 antibody (Cell Signaling Technology, #4370). Antibodies were diluted 1:1000 in 5% BSA in Tris-buffered saline with 0.1% Tween20, incubated overnight at 4 °C, and then probed for one hour at room temperature with a 1:3000 dilution of anti-rabbit (Cell Signaling Technology, #7074) IgG HRP-linked antibody and developed using Pierce ECL Western Blotting Substrate (ThermoFisher Scientific).

### 2.7. Statistical Analysis

All statistical analyses were performed using GraphPad Prism, version 9.2.0 (GraphPad Software, Inc., San Diego, CA, USA). The expression data for the selected ETS factors are presented as the mean ± standard deviation from at least three independent experiments. Statistical differences between cell lines with ETS factor silencing against scrambled control lines and between the DMSO and drug-treated samples were analyzed by unpaired students *t*-tests. Schematic figures were generated using BioRender software and images were exported under a paid subscription.

## 3. Results

### 3.1. ETS Factors Show a Wide Range of Expression in Thyroid Cancer Specimens but Remain Comparable across Thyroid Cancer Types

To assess which ETS factors are relevant in the control of *TERT* promoter in thyroid cancer specimens, we adopted an unbiased approach, evaluating their reported binding preferences and expression patterns in thyroid tumors and cell lines (Figure 1). We first compared the known consensus binding sequences for each of the 27 factors in the ETS family [25] against the de novo sites created by TPMs in the gene promoter (5′-CCGGAA-3′; underlined adenines are created by G > A mutations (reverse strand) at *TERT*-124 and-146). ETS factors from classes III and IV showed binding preferences (5′-GGA-3′ and 5′-GGAT-3′, respectively) that likely render them less dependent on de novo sites created by TPMs (Figure 1, top). 

We then leveraged transcriptomic data from The Cancer Genome Atlas (TCGA) analysis of PTC [9] as well as our own evaluation of PDTC, ATC, and thyroid cancer cell lines [8,20]. As shown in Figure 1, seven ETS factors, including FEV, ETV2, ELF5, and all from classes III (SPI1, SPIB, and SPIC) and IV (SPDEF), were not expressed in thyroid specimens, ruling them out from having any role in *TERT* promoter control in thyroid cells. Conversely, factors such as ETS1 and ETV5 showed high and consistent expression across thyroid cancer types, whereas most ETS proteins, including GABPA, were expressed at intermediate levels. Overall, these analyses narrowed the list of candidate ETS factors regulating *TERT* transcription in thyroid tumors down to 20 members.

### 3.2. Screening of 20 ETS in Cell Lines Shows Vvariable Effects on TERT Expression

Taking the list of 20 ETS factors expressed in thyroid cell lines and tumors, we employed short-hairpin RNAs (shRNA) targeting each factor individually and generated knocked-down stable cell lines (Figure 2A and Table 1). Initially, four thyroid cancer cell lines were employed: TCO1, SW1736, and C643, which all harbor *TERT* c.-124C>T mutation, and Cal62, which was used as a control for a wild-type *TERT* promoter. In these cell lines, stable knocking down of specific ETS factors had a variable effect on *TERT* expression when compared with scrambled shRNA cells (Figure 2B, left panel, and Appendix A). We reasoned that ETS factors that are specifically recruited to the de novo ETS site formed by the -124C>T mutation would modify *TERT* expression in TCO1, SW1736, and C643 cells, while Cal62 cells would remain unchanged. However, this pattern was not observed. Indeed, multiple factors exerted an effect on TERT transcription in a manner that was not always evidently TPM-mediated. Indeed, ELK1, ELK3, ELK4, ELF3, ELF4, ERF, and ETV1 knockdown resulted in significant reductions in *TERT* expression in at least two out of three *TERT*-mutant cell lines. Of those, ELK1, ELK3, ELF3, and ERF knockdown also induced at least 45% reductions in TERT expression in the Cal62 cell line (TERT wildtype). To further study the effects of these factors in an extended panel of cell lines, we selected the aforementioned seven ETS proteins, along with EHF and ETS2, which showed marginally significant reductions in *TERT* expression, and GABPA, which had been reported in other cancer lineages. Interestingly, in our experiments, GABPA knockdown produced more variable results and did not consistently result in a reduction in *TERT* levels as previously described upon short-term disruption to GABPA expression.

We subsequently screened the selected 10 ETS factors that showed considerable reductions in *TERT* expression in three additional cell lines: BHT101, BCPAP, and T238 (Figure 2B, right panel). From these data, knockdown of ETV1, ELK1, ELF3, and ERF consistently resulted in downregulation of the *TERT* expression across all or most cell lines appraised, with varying degrees of magnitude. In many cases where knockdown of an ETS factor resulted in a downregulation of *TERT* expression, a correlating reduction in *TERT* was observed in the Cal62 control cell line. Therefore, these observations may not represent factors that specifically regulate the mutant *TERT* promoter.

Of note, we confirmed the functional knockdown of ETS factors at the RNA level in these same specimens (Figure 2C and Appendix A). In 85% of these stable cell lines, there was at least a 50% reduction in expression of the targeted ETS factor compared to the scrambled shRNA control. Knockdown by shRNA was notably less effective in SW1736 cell line for unknown reasons.

Overall, unlike in other lineages, the systematic knockdown of ETS proteins specifically expressed in thyroid cell lines and tumors did not point to control of the mutant *TERT* promoter by one or more specific ETS factors. The former suggests a remarkable complexity of ETS-mediated regulation of *TERT* promoter and points to potential mechanisms of transcription factor promiscuity and/or compensation beyond those already reported.

### 3.3. GABPA Knockdown Does Not Affect TERT Expression across Thyroid Cancer Cell Lines Carrying TPMs

Given the prior body of evidence that pointed towards GABPA being a key regulator of mutant *TERT* promoter in glioblastoma, the effect of GABPA knockdown on *TERT* expression was further tested across a comprehensive range of thyroid cell lines harboring wild-type *TERT* promoter sequences and heterozygous or homozygous TPMs. In this analysis, the U251 glioblastoma cell line was also included as a positive control. GABPA was silenced using short-hairpin RNAs and cells were cultured for at least two weeks in selection media to observe the effect of sustained knockdown. Overall, GABPA knockdown did not show a consistent genotype-dependent effect on *TERT* expression across cell lines. The wild-type *TERT* promoter cell line Cal62 demonstrated a significant decrease in *TERT* expression upon GABPA knockdown against the paired scrambled shRNA control, whereas MDAT41 and CUTC5, also wild-type for TPMs, did not (Figure 3A). In comparison, the cell lines that are heterozygous for a *TERT* promoter mutation did not show a significant reduction in *TERT* expression (Figure 3B). In TCO-1 and Hth7 cells, which are homozygous for the *TERT* promoter mutations -124C>T and -146C>T, respectively, and which should theoretically facilitate GABPA binding, showed no effects on *TERT* expression upon GABPA knockdown (Figure 3C). This was not a result of poor knockdown efficiency as TCO-1 and Hth7 demonstrated sustained and consistent reduction of GABPA expressions (average decreases of 61.5% and 46.8%, respectively; Figure 3C). In contrast, continued GABPA knockdown in U251 glioblastoma cells confirmed the previously shown significant reduction in *TERT* expression (Figure 3D). 

Taken together, these data caution against GABPA being a universal regulator of the mutant *TERT* promoter in thyroid cancer cells and suggest that targeting GABPA over a sustained period is unlikely to result in a consistent reduction in *TERT* expression in this tumor type. 

### 3.4. Inhibiting MAPK Signaling Primarily Affects Expression of TERT in Cells with Wild-Type TERT Promoter

Considering the known role of MAPK signaling in regulating wild-type *TERT* expression, we sought to explore the effect of MAPK blockade in thyroid cancer cells with or without TPMs and to assess whether it cooperates with ETS to control *TERT* transcription. To this end, we treated cells with FDA-approved dabrafenib (RAF inhibitor) and/or trametinib (MEK inhibitor), depending on their driver mutation (BRAF^V600E^ or oncogenic RAS). To ensure accurate comparisons across cell lines with variable MAPK outputs, IC_50_ values for trametinib and dabrafenib were first determined, and each cell line was treated at its IC_50_ (Figure 4A). Cell lines were transduced with short-hairpin RNAs targeting the ETS factors—ETV1, ELK1, and GABPA—and cultured under puromycin selection media for at least two weeks prior to treatment to ensure sustained RNA knockdown (Figure 4A).

After treatment at the IC_50_ dose with the appropriate MAPK inhibitors, Cal62 and MDAT41 cells demonstrated a strong reduction in *TERT* expression across all shRNA conditions. Comparatively, BHT101 and BCPAPs were refractory to this treatment and did not show significant reductions in *TERT* expression at their IC_50_ doses. Although this may point to a partial switch of *TERT* regulation away from MAPK towards other transcription factors such as ETS factors in TPM-harboring cells, *TERT* expression was not further reduced upon knockdown of either ETV1, ELK1, or GABPA in BHT101 or BCPAP cells. Functional knockdown of the MAPK pathway was confirmed after the short-term treatment, as shown by diminished phospho-ERK1/2 levels (Figure 4C). Overall, our experiments did not show cooperativity between ETS- and MAPK-mediated control of *TERT* transcription but pointed to a shift in *TERT* mutant promoter regulation. 

## 4. Discussion

In this study, we comprehensively assessed the ability of the ETS family of transcription factors to control *TERT* transcription in thyroid cancer cells with and without TPMs. We also evaluated the *TERT* promoter genotype-dependent role of MAPK signaling in *TERT* expression in these same specimens. Our findings suggest a multi-layered pattern of regulation, likely more complex than that described in glioblastoma cells carrying TPMs. In our cell systems, multiple ETS factors regulated *TERT* expression in a manner that is not exclusive of the presence of TPMs. We also observed that TPMs likely render thyroid cancer cells less dependent to MAPK-mediated control of *TERT* transcription, whereas pharmacological inhibition of MAPK signaling effectively suppressed *TERT* mRNA levels in the absence of TPMs.

The discovery of hotspot mutations in the *TERT* core promoter, first in metastatic melanomas and subsequently in aggressive tumors of multiple lineages, revitalized the interest in understanding the mechanisms of telomerase reactivation in cancer. The fact that the two non-overlapping c.-124C>T and c.-146C>T mutations create de novo consensus binding sites for the ETS family of transcription factors made them excellent candidates to be controlling TPM-mediated TERT re-expression (2–4). The former prompted to the search for which ETS factors bind mutant *TERT* promoter as a first step towards exploring the idea of targeting *TERT* mutant transcription.

Several studies, typically using siRNA silencing, reporter assays, and ChIP approaches, on glioblastoma cells carrying TPMs demonstrated that the ETS factor GABPA controls *TERT* mutant promoter expression. Bell and colleagues sequentially knocked-down thirteen ETS transcription factors in two glioblastoma lines and showed that GABPA selectively reduces *TERT* transcription in the presence of the c.-124C > T mutation. They extended their observations to three other TPM-carrying tumor types: melanomas, hepatocellular, and urothelial carcinomas [13]. This same group elegantly showed that GABPA acts as a tetramer with their beta-isoform (GABPB) and that GABPβ1L disruption reverses replicative immortality in glioblastoma cells with TPMs [15]. Interestingly, another group showed that glioblastoma cells carrying the alternative hotspot mutation (*TERT* c.-146C>T) are bound by ETS factors ETS1/2, which cooperate with non-canonical NF-kB signaling via p52 to reactivate mutant *TERT* [26]. They subsequently showed that GABPA is also able to upregulate mutant *TERT* in glioblastoma and melanoma cells, but it does so, at least partially, via long-range chromatin interactions [14]. Overall, GABPA seems a solid candidate to explore TPM-specific epigenetic blockade of *TERT* re-expression in glioblastoma, but its role in thyroid cancers remains uncertain. 

In this regard, the capacity of GABPA to bind to mutant *TERT* promoters in thyroid cancer was reported through ChIP assays on the K1 papillary thyroid carcinoma cell line [16]. The authors suggested that FOS cooperates with GABPA in *TERT* mutant control. Our results show that GABPA stable silencing does not universally reduce TERT transcription in *TERT*-mutant thyroid cancer cells and that it also controls *TERT* wildtype promoter (e.g., in Cal62 cells). In addition, GABP seems an unattractive target for inhibition in thyroid cancers due to its recently reported tumor suppressor role [19]. 

Our study shows that multiple ETS factors are able to control *TERT* mutant transcription, suggesting that no clear (GABPA-like) candidate exists in this lineage and that compensatory mechanisms across ETS proteins likely operate. Our work confirms some of the observations and further expands the findings reported by other thyroid cancer groups. Of note, Bullock and colleagues nicely showed that both ELK1 and ETV5 are able to bind *TERT* promoter in discrete thyroid cell lines and that they do so in cooperation with thyroid transcription factor FOXE1 [17,18]. A separate study reported that ETV1, ETV4, and ETV5 preferentially bind *TERT* in various thyroid cell lines carrying c.-124C>T mutation [11]. Taken together, our exhaustive screening of all 20 ETS proteins expressed in thyroid cells show that no unequivocal ETS factor is responsible for *TERT* mutant control. We show that sustained knockdown of specific ETS over weeks, at most, moderately impacts *TERT* levels. This is a key difference from other studies, which assessed *TERT* knockdown transiently, not allowing for the likely compensatory mechanisms that other factors exert in our cells and probably in actual tumor specimens re-expressing telomerase. 

Another aspect of *TERT* promoter regulation that remains elusive is determining whether some of the reported non-canonical TPMs, which occur at very low frequencies (1–2% of cases) but also create de novo ETS sites, reactivate telomerase expression through the same mechanisms operating in *TERT* c.-124C>T or c.-146C>T specimens [27,28]. In this regard, it is reasonable to speculate that some of these rare TPMs might activate *TERT* to a different degree, while others could be passenger alterations resulting from an overdetailed dissection of this particular genomic *locus*. In any case, once the mechanisms of hotspot TPM-mediated expression of *TERT* are identified, it would be advisable to study these other non-coding lesions.

Our work also explored the role of a MAPK pathway blockade via pharmacological inhibition by dabrafenib and/or trametinib in *TERT* transcription. We observed that, at low doses (low nanomolar range), MAPK inhibition was very effective at suppressing *TERT* mRNA levels in cells without TPMs but not in their *TERT*-mutant counterparts. Although we acknowledge the preliminary nature of this observation, our results point to a potential partial switch of *TERT* mutant regulation, which might have been missed by other works employing MAPK inhibitors in the micromolar range, which fully suppressed this pathway [16,18]. Studying the cooperation between ETS- and MAPK-mediated regulation in *TERT* transcription is inherently difficult due to their roles in the absence of TPMs. This is because the *TERT* wildtype promoter has native ETS sites as well as binding motifs for Sp1 and c-Myc, both MAPK-activated factors that control *TERT* expression [20,21,22,23]. In addition, MAPK signaling is able to activate discrete ETS factors either via transcription [29] or phosphorylation [30], which would feed into both wildtype and mutant *TERT* regulation. However, in the absence of unambiguous ETS proteins controlling *TERT* mutant expression, the TPM-specific role of MAPK input remains elusive.

Beyond the regulation by the ETS/MAPK axis, *TERT* promoter has been shown to be subjected to other regulatory inputs, some of which might be modified by the presence of TPMs, whereas others likely operate as determinants of *TERT* baseline transcriptional repression. Although this is beyond the scope of this paper, the role of genomic insulator CTCF, which likely determines long-distance interactions [31] as well as the contribution of DNA methylation and allele-specific histone marks, which were recently characterized in thyroid cancer cell lines [18,32,33], are worth noting. Overall, these add to the idea of a tightly regulated, multi-faceted control of *TERT* transcription, which only becomes unchecked during cancer transformation. Incidentally, it is possible that some of the variability in ETS-binding is determined by specific co-factors and chromatin states that might respond to other signaling inputs. Other differences are likely attributable to the stable nature of the ETS silencing used here (vs. transient approaches in other studies), which likely allows cancer cells to enhance compensatory mechanisms. Overall, we believe that our observations, together with the remarkable efforts from other colleagues, have exhausted the possibilities of finding unequivocal, mutation-specific, ETS factors controlling *TERT* in thyroid cancers, and we caution against the generalization of discrete ETS findings from other tumor lineages. In the future, a less ETS-centric vision of *TERT* promoter control will be required to fully understand this process. In this regard, our ongoing research directions involve exploring ways of assessing the proteomic landscape of this genomic region in defined genetic states using isogenic models.

## 5. Conclusions

In conclusion, our data suggest that the ETS-mediated regulation of *TERT* mutant promoter in thyroid cancers likely differs from that reported in other tumors carrying TPMs in the same hotspots. Epigenetic targeting of mutant *TERT* transcription remains an attractive yet ambitious goal due to the high prevalence of TPMs in advanced tumors and the cancer-specific nature of this approach. However, we believe that the former will first require a detailed understanding of the transcriptional regulation of *TERT* mutant promoter specifically in thyroid specimens. To achieve this, a precise role of the mechanisms of compensation across different ETS factors, the integration with other inputs (e.g., other regulators activated by MAPK and/or other pathways) and ultimately a factor-agnostic approach, ideally in cancer isogenic models, which assesses the epigenetic reconfiguration around *TERT* mutant promoter, will be required.

## Figures and Tables

**Figure 1 cancers-14-00357-f001:**
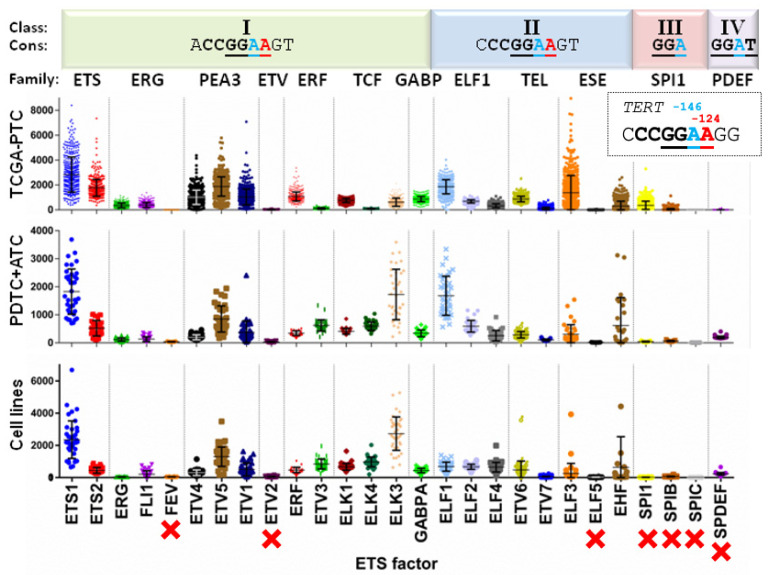
General classification and expression patterns of the ETS family of transcription factors in thyroid cancer specimens. ETS factors are divided in four (I–IV) classes (**top row**) based on their binding sequence preferences (**second row**). Consensus (“Cons”) sites are displayed, with core sequences shown in bold and the de novo ETS binding sites created by *TERT* promoter mutations at −124 and −146 highlighted in red and blue font, respectively (both C>T transitions, shown in the reverse strand in the dotted line box). The third row lists the names of the 12 ETS families, based on the presence of other protein domains. Graphs show the normalized expression levels from published thyroid cancer datasets, including PTC from the TCGA study (**top panel**) [9], and our data on PDTC + ATC (**middle**) [8] and thyroid cancer cell lines (**bottom**) [24]. The seven ETS factors not expressed in thyroid cancers are indicated with red crosses. Abbreviations: ETS = E26 transformation specific; TCGA = The Cancer Genome Atlas; PTC = Papillary Thyroid Cancer; PDTC = Poorly Differentiated Thyroid Cancer; ATC = Anaplastic Thyroid Cancer.

**Figure 2 cancers-14-00357-f002:**
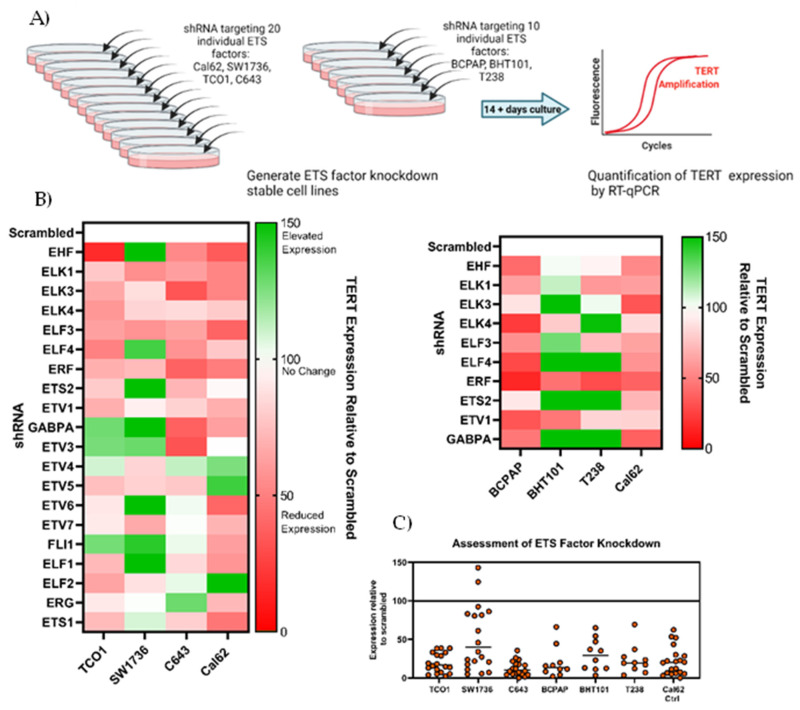
(**A**) The experimental design was setup such that twenty ETS factors were targeted by shRNA in four cell lines: TCO1, SW1736, C643, and Cal62. Ten ETS factors were targeted by shRNA in three further cell lines: BCPAP, BHT101, and T238. *TERT* expression was quantified by TaqMan qPCR. (**B**) Heatmap demonstrating relative changes in *TERT* expression compared with cells with shRNA targeting scrambled control in all seven cell lines. (**C**) Targeting of individual ETS factors by shRNA resulted in 85% cell lines with more than 50% reduction in expression of the targeted ETS factor.

**Figure 3 cancers-14-00357-f003:**
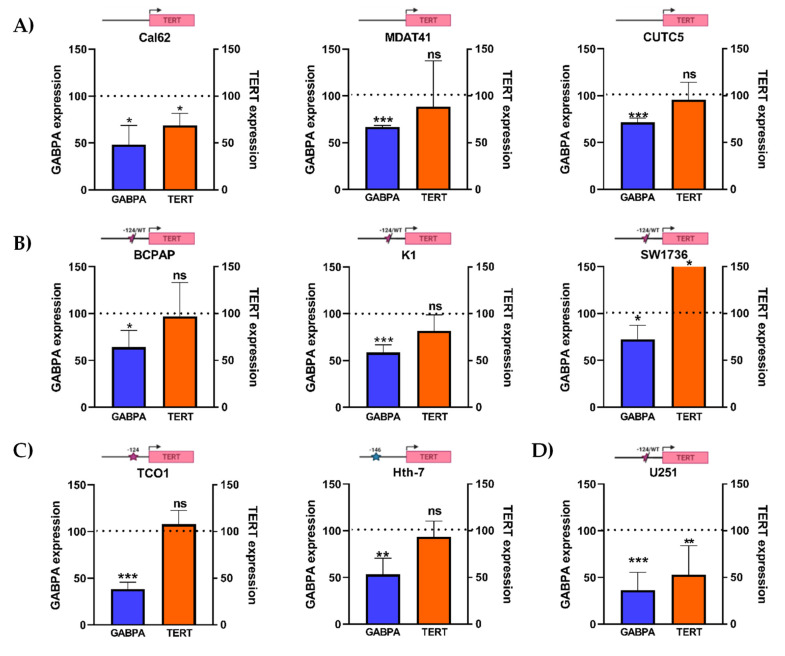
Stable knockdown of ETS factor GABPA did not result in consistent changes to *TERT* expression across multiple thyroid cancer cell lines. (**A**) Cell lines with wild-type *TERT* promoter sequences. (**B**) Cell lines with heterozygous -124C>T *TERT* promoter mutation. (**C**) Cell lines with homozygous -124C>T or -146C>T *TERT* promoter mutations and (**D**) U251 glioblastoma cell line, which is heterozygous for the -124C>T mutation. Across all cell lines, TERT expression was analyzed by RT-qPCR and then normalized to actin expression. Cal62, which acted as a non-*TERT* promoter mutation control, and U251 glioblastoma cells demonstrated a significant reduction in *TERT* expression. All other cell lines did not exhibit a significant reduction in *TERT* expression. Cells were cultured for at least two weeks prior to RNA extraction to appraise the longer-term effects of GABPA knockdown on *TERT* expression. All results are derived from at least triplicate experiments; ns = non-significant; * = *p*< 0.05; ** = *p* < 0.005, *** = *p* < 0.001.

**Figure 4 cancers-14-00357-f004:**
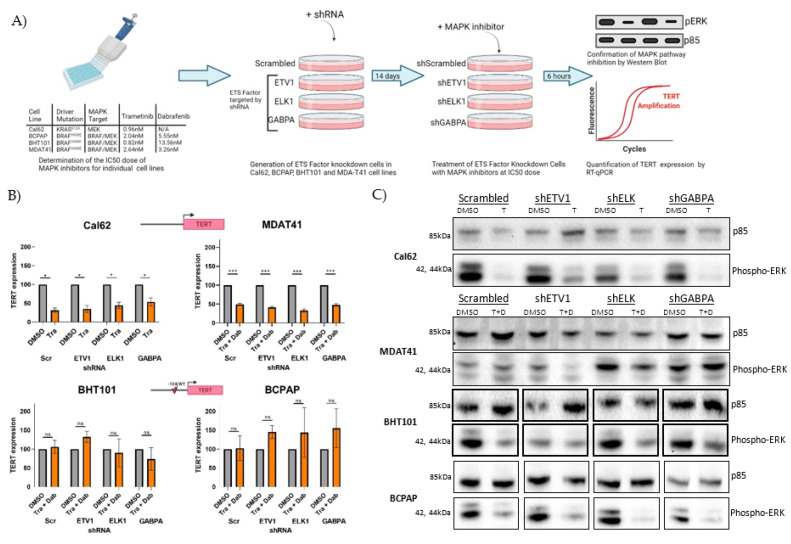
(**A**) The IC_50_ value for Trametinib (Cal62, 0.96 nM; BHT101, 2.04 nM; and BCPAP, 0.82 nM) and Dabrafenib (BHT101, 5.55 nM; BCPAP, 13.56 nM) were first calculated by cell viability assay on parental cell lines. Knockdown cell lines were generated using shRNA targeting ETV1, ELK1, and GABPA. shRNA cell lines were treated at the IC_50_ dose; then, protein and RNA were extracted to confirm pathway inhibition and to assess the effect on *TERT* expression, respectively. Created with Biorender.com (**B**) RT-qPCR analysis demonstrated a significant reduction in *TERT* expression across all Cal62 and MDAT41 ETS knockdown cell lines. BHT101 and BCPAP did not demonstrate a reduction in *TERT* expression after treatment with Trametinib and Dabrafenib for six hours. (**C**) Western blotting confirmed functional inhibition of the MAPK pathway after treatment with Trametinib or Trametinib + Dabrafenib using phospho-ERK as a biomarker. Samples were run on one membrane per cell line, and BCPAP and BHT101 images were arranged for a consistent loading order. ns = non-significant; * = *p*< 0.05; *** = *p* < 0.001.

**Table 1 cancers-14-00357-t001:** Name, genetic background, and other features of the cancer cell lines used in this study.

Cell Line	Tissue Derivation	MAPK Driver	*TERT* Promoter	Use
TCO1	Thyroid, ATC	BRAF V600E	c.-124C>T, HOM	1st screen, 20 ETS
SW1736	Thyroid, ATC	BRAF V600E	c.-124C>T, HET	1st screen, 20 ETS
C643	Thyroid, ATC	HRAS G13R	c.-124C>T, HET	1st screen, 20 ETS
Cal62	Thyroid, ATC	KRAS G12R	Wild type	1st screen, 20 ETS + MAPKi experiments
BCPAP	Thyroid, PTC	BRAF V600E	c.-124C>T, HET	2nd screen, 10 ETS + MAPKi experiments
BHT101	Thyroid, ATC	BRAF V600E	c.-124C>T, HET	2nd screen, 10 ETS + MAPKi experiments
T238	Thyroid, ATC	BRAF V600E	c.-124C>T, HET	2nd screen, 10 ETS
MDAT41	Thyroid, PTC	BRAF V600E	Wild type	Extended screen, GABPA + MAPKi experiments
CUTC5	Thyroid, PTC	BRAF V600E	Wild type	Extended screen, GABPA
K1	Thyroid, PTC	BRAF V600E	c.-124C>T, HET	Extended screen, GABPA
Hth7	Thyroid, ATC	NRAS Q61L	c.-146C>T, HOM	Extended screen, GABPA
U251	Glioblastoma	Unknown	c.-124C>T, HOM	Extended screen, GABPA

Abbreviations: PTC = Papillary Thyroid Cancer; ATC = Anaplastic Thyroid Cancer; HET = Heterozygous; HOM = Homozygous; ETS= E26 transformation specific; MAPK = Mitogen-activated protein kinase; MAPKi = MAPK inhibition.

## Data Availability

This manuscript includes gene expression analyses performed on openly available genomic data repositories, including the TCGA study of papillary thyroid carcinoma (http://firebrowse.org/?cohort=THCA&download_dialog=true, accessed on 1 January 2018) and the Gene Expression Omnibus dataset #GSE76039 (https://www.ncbi.nlm.nih.gov/geo/query/acc.cgi?acc=GSE76039 accessed on 1 January 2018); previously generated by us [8,24]).

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
