# Peer review of "Multiple ETS Factors Participate in the Transcriptional Control of TERT Mutant Promoter in Thyroid Cancers"

_cancers, 2022, doi:10.3390/cancers14020357_

Round 1
Reviewer 1 Report
The manuscript by Thornton et al. is a pertinent and well written work. The Authors describe the effects of ETS effectors silencing on TERT gene expression in different thyroid cancer cell lines using specific shRNAs.
The story builds on prior works by other groups in other tumors, demonstrating that ETS transcription factors gain transcriptional activity through de novo binding sites generated by mutations of TERT promoter region. This study now extends our knowledge to a novel ETS-mediated regulation of TERT mutant promoter in thyroid cancer. Here, the Authors speculate that no unequivocal candidate exists to control TERT mutant expression, but multiple ETS factors are able to control the expression of TERT mutant in different way. In addition, they investigated the effects of MAPK inhibitors in thyroid cancer cell lines WT or TERT mutated under control of specific ETS silencing, showing no cooperativity between ETS and MAPK-mediated control of TERT transcription.
Overall, I found the paper and findings to be compelling. The figures are nicely done but, few of them, in my opinion, need some adjustment.
I have some MINOR comments:
- All gene symbol names should be reported in Italic, please double check in the text.
- Results, paragraph 3.2: based on what the authors report in this paragraph, 7 ETS (ELK1, ELK3, ELK4, ELF3, ELF4, ERF and ETV1) knockdown resulted in a significant reduction of TERT expression in at least 2 out of 3 cell lines (I assume TCO1 and C643) and of these 7 only 3 (ELK4, ELF4 and ETV1) did not affect TERT expression in WT cells but, if I look at the figure 2B left, seems that none of them clearly effect the expression of TERT in Cal62 cells. Please, clarify.
- Figure 2B right, 10 ETS factors that showed considerable reduction on TERT expression were screened in three additional cell lines. 3 out of 10 knockdown ETS (ETV1, ELK1, and ERF) resulted in a downregulation of TERT in most cell lines with varying degrees of magnitude. What about ELF3? It looks like to have the same trend of ELK1. Please, clarify.
- Line 237, I assume that it should be “Figure 2B, right panel” and not 2C. If yes, please, change in the text.
- Supplemental figure 1, it would be appreciated if in all 4 histograms the order of the genes were the same, please, double check.
- Table 1, please double check the abbreviations.
- Figure 3, please add the alphabetical partition in the legend.
- Based on what did the Authors choose the 3 ETS factors ETV1, ELK1 and GABPA to investigate the effects of MAPK inhibitors on TERT expression?
- Recently new TERT promoter mutations were reported in literature. These alterations were detected in PTC, ATC and MTC (none of them are my papers). These new TERT promoter mutations should stimulate the transcription activity through the generation of de novo consensus binding sites for ETS family transcription factors. What do the Authors think about it? It would be useful for the Authors to add a brief paragraph in Discussion about these new findings and new potential studies.
- Do the Authors have any thoughts about the possibility to generate a mouse model TERT+ to test in vivo the effect of ETS silencing in future? Is it realistic?
Reviewer 2 Report
In this paper the authors analysed the effects of knockdown of 20 ETS transcription factors in human thyroid cancer cell lines. The genetic background of all the cell lines was known, in particular driver mutations and TERT promoter mutations are reported.
The study is interesting and sheds light on an important therapeutic issue, although caution is required before transferring these results to thyroid tumors, as authors themself assert.
Minor revision needed:
It would be important to show TERT expression data in the different cell lines. In fact, the TERT expression data of “parental” cell lines (cells infected with pLKO_shScramble) are reported only as a reference (and for this reason, the value is always 100) but in this way it is not possible to compare the TERT expression data between different cells lines.
In figure 1, the Authors showed ETS expression data for thyroid cell lines and refer to the bibliographic entry n°20 (Takakura M and others, Cancer Res. 1999;59(3):551-7). However, the cited paper analysed TERT promoter of cells derived from cervical cancers. Please, verify.
I suggest to add a clearer caption of the color code used in figure 2B. From the legend, it appears that the TERT expression is showed in white when not-affected by the silencing, in green when it is up-regulated and in red when it is down-regulated. Is it correct?
Were the IC50 values for each cell line calculated before or after transfection with shRNA? In materials and methods section and in the text (lines 306-310) appears that IC50 values were evaluated before transfecting cells but in figure 4A it seems the opposite. Please, clarify.
